# Assessing Pulmonary Epithelial Damage in Hantavirus Cardiopulmonary Syndrome: Challenging the Predominant Role of Vascular Endothelium through sRAGE as a Potential Biomarker

**DOI:** 10.3390/v15101995

**Published:** 2023-09-26

**Authors:** Gabriela Meza-Fuentes, René López, Cecilia Vial, Lina Jimena Cortes, Mauricio A. Retamal, Iris Delgado, Pablo Vial

**Affiliations:** 1Instituto de Ciencias e Innovación en Medicina, Facultad de Medicina, Clínica Alemana Universidad del Desarrollo, Av. Plaza #680, San Carlos de Apoquindo, Las Condes, Santiago 7610658, Chile; gmezaf@udd.cl (G.M.-F.); mcvial@udd.cl (C.V.); linacortes@udd.cl (L.J.C.); pvial@udd.cl (P.V.); 2Grupo Intensivo, ICIM, Facultad de Medicina, Clínica Alemana Universidad del Desarrollo, Santiago 7590943, Chile; 3Departamento de Paciente Crítico Clínica Alemana de Santiago, Santiago 7610658, Chile; 4Centro de Fisiología Celular e Integrativa, Facultad de Medicina, Clínica Alemana Universidad del Desarrollo, Santiago 7610658, Chile; mretamal@udd.cl; 5Centro de Epidemiología y Políticas de Salud, Facultad de Medicina, Clínica Alemana Universidad del Desarrollo, Santiago 7610658, Chile; idelgado@udd.cl

**Keywords:** hantavirus cardiopulmonary syndrome, acute respiratory distress syndrome, sRAGE

## Abstract

Hantavirus cardiopulmonary syndrome (HCPS) is a severe respiratory illness primarily associated with microvascular endothelial changes, particularly in the lungs. However, the role of the pulmonary epithelium in HCPS pathogenesis remains unclear. This study explores the potential of soluble Receptors for Advanced Glycation End-products (sRAGE) as a biomarker for assessing pulmonary epithelial damage in severe HCPS, challenging the prevailing view that endothelial dysfunction is the sole driver of this syndrome. We conducted a cross-sectional study on critically ill HCPS patients, categorizing them into mild HCPS, severe HCPS, and negative control groups. Plasma sRAGE levels were measured, revealing significant differences between the severe HCPS group and controls. Our findings suggest that sRAGE holds promise as an indicator of pulmonary epithelial injury in HCPS and may aid in tracking disease progression and guiding therapeutic strategies. This study brings clarity on the importance of investigating the pulmonary epithelium’s role in HCPS pathogenesis, offering potential avenues for enhanced diagnostic precision and support in this critical public health concern.

## 1. Introduction

In 1993, the first cases of hantavirus cardiopulmonary syndrome (HCPS), also called hantavirus pulmonary syndrome (HPS), were reported, exhibiting similarities to acute respiratory distress syndrome (ARDS) with a mortality rate of 75% [1]. In South America, the primary cause of HCPS is the Andes virus [2,3], which stands apart from other hantaviruses due to its ability to be transmitted from human to human with pandemic potential [4]. Hantavirus infection primarily targets the microvascular endothelium, with severe human disease linked to microvascular leakage [5]. The severe clinical impact of HCPS primarily affects the lungs, commonly presenting as the rapid development of non-cardiogenic diffuse pulmonary edema [6].

The precise mechanism underlying pulmonary vascular leakage during hantavirus infection and its association with HCPS remain incompletely understood. The progression to HCPS is believed to involve direct factors, such as viral replication, and indirect factors related to the immune response induced by the viral infection. Numerous studies have demonstrated that hantavirus infection can trigger changes in the vascular endothelium’s barrier function, activating both endothelial cells and leukocytes, thereby inducing the synthesis and release of proinflammatory cytokines [7,8,9].

Furthermore, insights from previous studies suggest that HCPS is associated with a serum cytokine profile that is consistent with potential immune responses within lung tissue, including the pulmonary epithelium [10]. This serum profile indicates robust activation of mononuclear immune effectors, including T lymphocytes, natural killer (NK) cells, and dendritic cells. These findings raise fundamental questions regarding the role of the pulmonary epithelium in the pathogenesis of HCPS and its interaction with the immune response.

However, the investigation into pulmonary epithelial damage in hantavirus infection remains insufficient. Currently, a specific biomarker of epithelial lung damage exists, known as the receptor for advanced glycation end products (RAGE). RAGE is prominently expressed in type 1 pneumocytes, and its soluble form (sRAGE) plasma levels have shown a correlation with pulmonary epithelial damage [11]. Jabaudon et al. discovered that plasma sRAGE levels were associated with altered alveolar fluid clearance in ARDS patients and animals with acute lung injury, cementing RAGE as a marker of pulmonary injury in humans [12]. On the other hand, there is a significantly greater difference in sRAGE levels in patients with non-focal ARDS compared to focal ARDS, even establishing a cut-off point of 1188 pg/mL with a sensitivity of 94% and a specificity of 84% [13]. Increased plasma levels of sRAGE have also been associated with poor clinical outcomes in ARDS patients ventilated with higher tidal volumes (VT), suggesting that mechanical ventilation with lower VT may be beneficial in decreasing lung epithelial injury [14].

Despite these prior findings, the relationship between plasma sRAGE levels and hantavirus infection remains unexplored. This could potentially provide insights into the pivotal role of epithelial tissue in HCPS development. Therefore, the primary objective of this study is to investigate sRAGE levels in patients suffering from HCPS caused by the Andes hantavirus. By addressing this question, we aim to expand our understanding of the underlying viral pathogenesis and potentially unveil innovative perspectives for identifying therapeutic targets against this significant public health threat.

## 2. Materials and Methods

### 2.1. Study Design and Patients

This study is an observational, retrospective, and analytical investigation conducted using a patient cohort derived from a database established by the Hantavirus and Zoonosis Program at the Instituto de Ciencias e Innovación en Medicina, Faculty of Medicine, Clínica Alemana-Universidad del Desarrollo.

The primary data used in this study were collected prospectively between 2017 and 2019 from a cohort of patients recruited for future research across seven healthcare facilities in six Chilean cities. The diagnosis of HCPS was confirmed in all cases through a quantitative enzyme-linked immunosorbent assay (ELISA) for the detection of ANDV-specific immunoglobulin M or RT-qPCR [15].

Demographic information, clinical details, laboratory results, and hospital mortality statistics were systematically gathered using a standardized case record form. The deidentified data were subsequently input into a dedicated database. Patients were classified into two different categories based on their clinical presentation, namely mild, moderate, and severe diseases, with the following definitions:

Mild disease: patients with early cardiopulmonary manifestations (dyspnea, chest X-ray opacities) without respiratory failure (defined as PaO_2_ < 60 mmHg at FiO_2_ 21%).

Severe disease: patients that required advanced life support (defined as the requirement of inotropic and/or ventilatory support).

Non-infected Intensive Care Unit (ICU) patients: intubated patients without respiratory failure or hantavirus infection.

The institutional ethics committee granted approval for this non-interventional study involving anonymized data and exempted the need for informed consent.

### 2.2. Eligible Patient Population

We conducted a demographic and laboratory assessment of all patients in our study. Our study enrolled critically ill patients who were admitted to ICUs across multiple healthcare facilities in Chile. The inclusion criteria for the two hantavirus-infected groups (severe and mild) encompassed enrollment in the study, a confirmed diagnosis of hantavirus infection, and being 15 years of age or older.

For the group of non-infected ICU patients, inclusion criteria included enrollment in the study and ICU admission.

Exclusion criteria were standardized across all three groups and encompassed the following conditions: acute diabetes exacerbation, renal replacement therapy for end-stage renal disease, Alzheimer’s disease, amyloidosis, metastatic solid tumors, hematologic neoplasms, acute exacerbation of chronic obstructive pulmonary disease (COPD), chronic liver disease (Child–Pugh C), lung transplantation, asthma, and patients on extracorporeal membrane oxygenation (ECMO). These criteria were applied due to previous reports indicating deregulated sRAGE levels in these specific conditions [13].

These stringent criteria were employed to refine the sample and generate a more focused dataset, ensuring the selection of patients who met the specific criteria outlined in our study.

### 2.3. Outcomes

The primary outcome was plasma sRAGE concentration (pg/mL). Secondary outcomes were platelet count, ventilatory mechanics, and oxygenation.

### 2.4. Procedure

Blood samples were obtained from patients with severe hantavirus and the negative control group on the first day of intubation. In the mild hantavirus group, blood samples were collected on the first day following diagnosis. These blood samples were then processed for sRAGE measurement using a commercially available enzyme-linked immunoassay (Human RAGE Quantikine ELISA; R&D Systems, Minneapolis, MN, USA).

### 2.5. Statistics

Statistical analysis was performed using Prism 9, version 9.4.1 (GraphPad Software, San Diego, CA, USA). Nonparametric data are presented as the median (interquartile range). To compare variables among different groups, the Kruskal–Wallis test was employed for non-normally distributed data, and the chi-square test was used for categorical variables. For plasma sRAGE levels in groups, we utilized the Kruskal–Wallis test. Subsequent post-hoc analyses were conducted employing Dunn’s multiple comparisons test. Correlation analysis was performed using Spearman’s coefficient. Statistical significance was defined as a *p*-value < 0.05.

## 3. Results

### 3.1. Baseline Characteristics of the Study Patients

The baseline characteristics of the study patients are presented in Table 1, revealing statistically significant differences among the severe hantavirus, mild hantavirus, and control groups in various parameters, including platelet count, SOFA score, arterial oxygen partial pressure to inspired oxygen fraction ratio P/F ratio, and plasma sRAGE concentration. These findings highlight the differences in the plasma sRAGE concentration as particularly noteworthy.

### 3.2. Comparison of sRAGE among HCPS Severity Groups

Significant differences were notably observed between the severe hantavirus group and non-infected ICU patients regarding plasma sRAGE concentration. The median concentration of sRAGE in the severe hantavirus group was 11,060 pg/mL (95% CI: 2590–15,800), while the median in the mild hantavirus group was 2960 pg/mL (95% CI: 1469–4552). In contrast, the median sRAGE concentration in non-infected ICU patients was 1414 pg/mL (95% CI: 756.3–2041) (Figure 1).

### 3.3. Correlation between Platelet Count and sRAGE Concentration

We conducted a Spearman correlation analysis to assess the relationship between the platelet count and the sRAGE concentration in both severe and mild hantavirus patient groups. The analysis revealed a strong negative correlation (Spearman’s r = −0.866), which was statistically significant (*p* = 0.004) (Figure 2). This indicates that as platelet count decreases, sRAGE concentration tends to increase in patients with hantavirus infection.

### 3.4. Correlation Analysis with Other Variables

In addition to the correlation analysis conducted with platelet count and sRAGE concentration, we performed correlation analyses between sRAGE concentration and all other variables presented in Table 1. However, no statistically significant correlations were observed with any of these variables.

## 4. Discussion

The findings of this study highlight the significance of sRAGE as a potential biomarker for assessing pulmonary epithelial damage in HCPS. Although preliminary, our investigation revealed substantial differences in plasma sRAGE concentrations among the severe hantavirus, mild hantavirus, and control groups, suggesting that sRAGE may hold promise in elucidating the intricate pathophysiology of HCPS.

While HCPS is primarily associated with microvascular endothelial damage in the lungs [16,17], the role of the pulmonary epithelium has been relatively underexplored. Our study challenges the prevailing notion that endothelial dysfunction is the sole driver of this syndrome. The substantial increase in sRAGE levels, particularly in the severe hantavirus group, suggests that pulmonary epithelial injury plays a significant role in HCPS pathogenesis, potentially disrupting alveolar clearance mechanisms and resulting in pulmonary epithelial damage, as observed in prior studies [12]. This disruption of alveolar clearance may contribute to the development of severe pulmonary edema and respiratory failure in HCPS patients, aligning with previous research demonstrating the correlation of sRAGE with lung epithelial damage in conditions such as ARDS [18,19], which shares similarities with the pathogenesis of HCPS.

Our Spearman correlation analysis unveiled a robust negative correlation between platelet count and sRAGE concentration in both severe and mild hantavirus patient groups. This correlation strongly reinforces the connection between pulmonary epithelial damage and disease severity. Previous studies have consistently shown that the progression from mild to severe hantavirus infection is directly associated with a low platelet count [20,21]. This further emphasizes the potential role of sRAGE as a valuable biomarker in assessing the continuum of disease severity in hantavirus infection.

While our study provides valuable insights, it is essential to acknowledge certain limitations. One notable consideration is the relatively modest sample size, which, while sufficient for the current analysis, may warrant caution when generalizing our findings to broader populations. To affirm the potential of sRAGE as a predictive and prognostic marker in HCPS, it is advisable to explore larger patient cohorts. Additionally, conducting longitudinal assessments could offer a deeper understanding of the dynamic changes in sRAGE levels throughout the course of the disease.

Moreover, our study has initiated an exploration into the role of the pulmonary epithelium in the pathogenesis of hantavirus infections. This emerging avenue of research presents promising prospects for further investigation. Expanding upon this groundwork, future studies in larger cohorts and longitudinal settings are imperative to comprehensively elucidate the clinical utility of sRAGE as a biomarker in HCPS and to explore the intricate mechanisms underpinning its relationship with disease severity.

## 5. Conclusions

In conclusion, our study illuminates the potential of sRAGE as a promising biomarker for assessing pulmonary epithelial damage in HCPS. While HCPS has traditionally been associated with microvascular endothelial damage, our findings challenge this conventional understanding by underscoring the significance of pulmonary epithelial injury in the syndrome’s pathogenesis.

The substantial differences in plasma sRAGE concentrations observed among severe hantavirus patients, mild hantavirus patients, and non-infected ICU patients suggest that sRAGE may hold considerable promise in elucidating the complex pathophysiology of HCPS.

Furthermore, the robust negative correlation between platelet count and sRAGE concentration in hantavirus-infected patients underscores the intimate link between pulmonary epithelial damage and disease severity. This emphasizes the potential of sRAGE as a valuable biomarker for assessing the continuum of disease severity in hantavirus infection.

In summary, our study not only enhances our understanding of HCPS but also opens doors to innovative approaches for identifying therapeutic targets against this critical public health concern, potentially leading to improved outcomes for HCPS-affected patients.

## Figures and Tables

**Figure 1 viruses-15-01995-f001:**
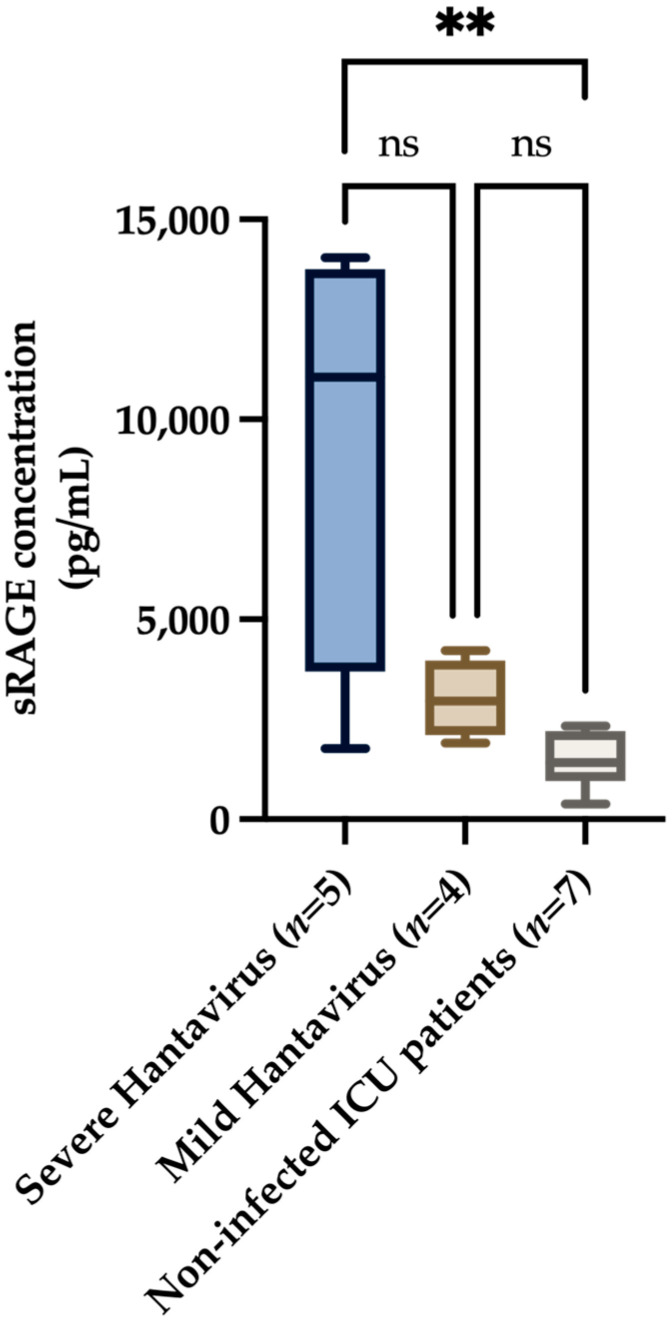
Plasma sRAGE concentration in the severe hantavirus group, the mild hantavirus group, and non-infected ICU patients. The data are presented as a graphical representation of the comparison between the groups. For plasma sRAGE level comparisons across different groups, Kruskall–Wallis was utilized. Dunn’s multiple comparisons test was conducted for post hoc analysis. A *p*-value of < 0.05 was considered statistically significant. ** = *p* value < 0.01; ns—non-significant.

**Figure 2 viruses-15-01995-f002:**
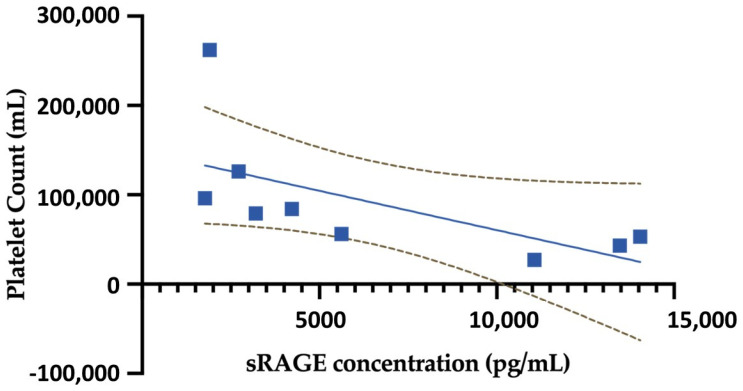
Correlation between platelet count and sRAGE concentration in hantavirus patients. This figure illustrates the results of a Spearman correlation analysis conducted to assess the relationship between platelet count and sRAGE concentration in both severe and mild hantavirus patient groups. A significant and strong negative correlation (Spearman’s r = −0.866, *p* = 0.004) was observed between these two variables in both hantavirus patient groups. The dashed line in the graph represents the 95% confidence interval of the simple linear regression, indicating the expected variability in the relationship between platelet count and sRAGE concentration in the studied population.

**Table 1 viruses-15-01995-t001:** Baseline characteristics of the study sample. (*N* = 16). P/F ratio (arterial oxygen partial pressure to inspired oxygen fraction ratio); MAP (mean arterial pressure); SOFA (sequential organ failure assessment); vd/vt (dead space fraction); sRAGE concentration (the concentration of soluble Receptors for Advanced Glycation End-products).

Characteristic	Mild Hantavirus (n = 4)	Severe Hantavirus (n = 5)	Non-Infected ICU Patients (n = 7)	*p* Value
Age, years, median (IQR)	38.5 (23–57)	22 (16–38)	40 (21–46)	0.363
Male, N (%)	4 (100)	3 (60)	3 (42.9)	0.168
Hematocrit, %, median (IQR)	42.75 (35.7–327.4)	46 (42.95–51.4)	39.3 (28.7–45)	0.067
Platelet count, mL, median (IQR)	105,000 (80,250–228,000)	53,000 (35,000–76,000)	262,000 (194,000–303,000)	0.004 **
P/F ratio, median (IQR)	410 (276–412)	190 (152.9–255.6)	396 (380–531)	0.011 *
MAP, mmHg, median (IQR)	83 (72.5–85.75)	80 (66–82.5)	85 (82–97)	0.052
SOFA, score, median (IQR)	0.33 (0.04–0.62)	10 (7.5–10)	1 (0–6)	0.006 **
Static Compliance, mL/cmH_2_O, median (IQR)	-	45.35 (32.75–52.93)	36 (33–56)	0.927
Driving Pressure, cmH_2_O, median (IQR)	-	9 (6.75–10.5)	9 (7–11)	0.92
vd/vt, median (IQR)	-	41.1 (36.75–55.05)	40.5 (29–42.9)	0.507
sRAGE concentration, pg/mL, median (IQR)	2960 (2105–3965)	11,060 (3697–13,760)	1414 (953–2207)	0.009 **

* *p* < 0.05 (level of significance). ** *p* < 0.01 (level of significance).

## Data Availability

Not applicable.

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
