# Peer review of "Assessing Pulmonary Epithelial Damage in Hantavirus Cardiopulmonary Syndrome: Challenging the Predominant Role of Vascular Endothelium through sRAGE as a Potential Biomarker"

_viruses, 2023, doi:10.3390/v15101995_

Round 1
Reviewer 1 Report
Comments and Suggestions for The Editor
This study, although preliminary, can open new field of investigation about HCPS pathology. The authors point to sRAGE as a promising and potential biomarker for assessing pulmonary epithelial damage in HCPS. Authors measured sRAGE and correlated it to the disease and to platelets.
The study is interesting, the paper is easy to read, although a limitation is represented by the low number of subjects, as specified by the authors in the discussion.
Only minor revisions are suggested.
- Can the authors discuss better in Introduction or Discussion the role and/or the involvement of RAGE in lung HCPS pathology or in the epithelial damage?
- Can the authors add some information about the level of inflammatory mediators (for example some cytokine) in HCPS serum patients? Are these information available in literature data?
Reviewer 2 Report
The investigation into pulmonary epithelial damage in hantavirus infection remains insufficient. The relationship between plasma sRAGE levels, a specific biomarker of epithelial lung damage, and hantavirus infection remains unexplored. The findings of this study highlight the significance of sRAGE as a potential biomarker for assessing pulmonary epithelial damage in HCPS. In my opinion, the topic is very interesting, and the authors made a nice contribution. Besides, the methodology and analysis of the data is accurate and appropriate, and the effects observed are interesting and relevant and were developed using adequate models.
Importantly, the authors conclude that their study illuminates the potential of sRAGE as a promising biomarker for assessing pulmonary epithelial damage in HCPSO. Blood samples were obtained from patients with severe hantavirus on the first day of intubation, while in the mild hantavirus group, blood samples were collected on the first day following diagnosis. I understand that this illness has three distinct phases: the prodrome, cardiopulmonary phase, and the convalescent phase. After a non-specific prodromal phase, patients have significant respiratory symptoms. Thus, considering that in those mild patients the samples were obtained the first day after diagnosis (during the prodrome pahse), have the authors also obtained samples from the severe hantavirus patients the first day after diagnosis ( during the prodrome phase)? In that case, I think that the authors could compare these samples with those obtained the first day after intubation in the severe hantavirus patients, in order to analyze whether it is an indicator of the underlying pathogenic process responsible for disease progression.
